# Recruitment Kinetics of XRCC1 and RNF8 Following MeV Proton and α-Particle Micro-Irradiation

**DOI:** 10.3390/biology12070921

**Published:** 2023-06-27

**Authors:** Giovanna Muggiolu, Eva Torfeh, Marina Simon, Guillaume Devès, Hervé Seznec, Philippe Barberet

**Affiliations:** University Bordeaux, CNRS, LP2I, UMR 5797, 33170 Gradignan, Franceguillaume.deves@cnrs.fr (G.D.); herve.seznec@cnrs.fr (H.S.)

**Keywords:** charged-particle microbeam, DNA repair, time-lapse microscopy, recruitment kinetics

## Abstract

**Simple Summary:**

We used the charged-particle microbeam installed at the AIFIRA facility to perform micro-irradiation experiments and measure the recruitment kinetics of DNA signaling and repair proteins. We developed and validated image acquisition and processing methods to enable a systematic study of the recruitment kinetics of two GFP-tagged proteins (GFP-XRCC1 and GFP-RNF8) after irradiation with protons and α-particles. Online measurement of fluorescence intensity and recruitment time as a function of particle type and number allowed us to characterize the differences in behavior between the two proteins.

**Abstract:**

Time-lapse fluorescence imaging coupled to micro-irradiation devices provides information on the kinetics of DNA repair protein accumulation, from a few seconds to several minutes after irradiation. Charged-particle microbeams are valuable tools for such studies since they provide a way to selectively irradiate micrometric areas within a cell nucleus, control the dose and the micro-dosimetric quantities by means of advanced detection systems and Monte Carlo simulations and monitor the early cell response by means of beamline microscopy. We used the charged-particle microbeam installed at the AIFIRA facility to perform micro-irradiation experiments and measure the recruitment kinetics of two proteins involved in DNA signaling and repair pathways following exposure to protons and α-particles. We developed and validated image acquisition and processing methods to enable a systematic study of the recruitment kinetics of GFP-XRCC1 and GFP-RNF8. We show that XRCC1 is recruited to DNA damage sites a few seconds after irradiation as a function of the total deposited energy and quite independently of the particle LET. RNF8 is recruited to DNA damage sites a few minutes after irradiation and its recruitment kinetics depends on the particle LET.

## 1. Introduction

The harmful effects of ionizing radiations can be attributed to the damage of a cellular target, usually identified as nuclear DNA, via direct absorption of radiation energy [1,2]. Irradiation leads to complex lesions consisting of clusters of damaged bases, single-strand breaks (SSBs), and double-strand breaks (DSBs) [3]. Cells have developed efficient defense mechanisms to repair these lesions based on several repair pathways involving various proteins. These mechanisms can be studied in cellulo and in situ to understand the role and recruitment of the signaling and repair proteins by visualizing radiation-induced foci (IRIF). Time-lapse fluorescence imaging coupled to irradiation devices provides information on the kinetics of protein accumulation, from a few seconds to several minutes after irradiation [4]. Several methods have been developed to trigger localized DNA lesions and analyze subsequent cellular responses in real-time in living cells, immediately after irradiation. These include laser-based micro-irradiation setups [5,6,7,8,9] and ionizing radiations [10,11,12,13,14,15]. In contrast with the photon irradiation offered by lasers, which can generate a wide variety of lesions depending on photon wavelength, energy, and exposure time [16], charged-particle provide more well-behaved energy deposition equally distributed over all molecular species [11,17]. The selection of the charged-particle type and energy provides a way to vary the density of ionizations at the sub-micrometer scale depending on their linear energy transfer (LET). This physical quantity is related to the track structure and characterizes the damage complexity. Densely ionizing radiation, with a high LET (e.g., α-particles), produces more complex lesions than those produced by radiations with a lower LET [18], which causes significantly sparser damage. Charged-particle microbeams allow the choice of the incident particle type and energy as well as the number of particles delivered in micrometric areas inside the cell nucleus [14]. This unique feature enables modification of the micrometer- and nanometer-scale distributions of the energy deposited by ionizing radiations and thus the density of DNA damage. In addition to irradiations performed experimentally, the energy deposits as well as the spatial distribution of ionization and excitation events in the target can be calculated using Monte Carlo codes such as Geant4 and its low energy extension Geant4-DNA [19,20]. Therefore, the biological response can be related to the physical dose deposited at the micrometric scale.

We report the use of a microbeam installed at the AIFIRA facility (Bordeaux, France) [21] to selectively irradiate micrometric areas in cell nuclei with protons and α-particles and measure the recruitment kinetics of the XRCC1 (X-Ray Repair Cross Complementing 1) and E3 ubiquitin-protein ligase (RNF8) proteins using online fluorescence time-lapse microscopy. We used the corresponding GFP-tagged proteins to validate our ability to target precisely cell nuclei, evaluate their behavior and characterize the early events using online microscopy [15,22]. In this work, we investigate their recruitment kinetics as a function of the dose and LET by irradiating with increasing numbers of 3 MeV protons (H^+^, 12 keV·µm^−1^) or α-particles (He^+^, 148 keV·µm^−1^) focused on a micrometer beam spot. Briefly, XRCC1 is a scaffold protein involved in DNA single-strand break repair by mediating the assembly of DNA break repair protein complexes involved in the efficient repair of DNA single-strand breaks formed by exposure to ionizing radiation and alkylating agents. This protein interacts with multiple enzymatic components of DNA single-strand break repair (SSBR) including DNA kinase, DNA phosphatase, DNA polymerase, DNA deadenylase, and DNA ligase activities that collectively are capable of accelerating the repair of a broad range of DNA single-strand breaks (SSBs) [7,23,24,25,26]. Furthermore, XRCC1 plays a crucial role in directly facilitating the nuclear localization of LIG3, expanding its function beyond Base Excision Repair (BER). Notably, XRCC1 actively participates in an error-prone Double-Strand Break Repair process known as alternative end joining (Alt-EJ) or alternative NHEJ. Unlike the traditional NHEJ pathway, Alt-EJ operates independently of NHEJ proteins and instead relies on base excision/single-strand break repair proteins, including PARP1, XRCC1, and DNA ligase 1 or 3 (LIG1/3), as well as XRCC1, for the joining of Double-Strand Break (DSB) termini [27,28,29,30]. The E3 ubiquitin-protein ligase (RNF8) plays a key role in DNA damage signaling via two distinct roles: (i) by mediating the Lys-63-linked ubiquitination of histones H2A and H2AX and promoting the recruitment of DNA repair proteins at double-strand breaks (DSBs) sites, and (ii) by catalyzing Lys-48-linked ubiquitination to remove target proteins from DNA damage sites. Following DNA DSBs, it is recruited to the sites of damage by ATM-phosphorylated MDC1 and catalyzes the Lys-63-linked ubiquitination of histones H2A and H2AX, thereby promoting the formation of TP53BP1 and BRCA1 ionizing radiation-induced foci (IRIF). Following DNA damage, RNF8 mediates the ubiquitination and degradation of POLD4/p12, a subunit of DNA polymerase delta. In the absence of POLD4, the DNA polymerase delta complex exhibits higher proofreading activity [9,31,32,33,34,35]. We used two GFP-tagged proteins as biological indicators to illustrate our ability to micro-irradiate and analyze cell nucleus responses in real-time. These proteins, XRCC1 and RNF8, have a well-established importance in assessing temporal and spatial recruitment in living cells. Multiple versions of fluorescently-tagged XRCC1 and RNF8 have been designed and studied using a variety of techniques such as laser micro-irradiation, soft X-rays, and charged particle microbeams [9,25,36,37,38,39,40,41,42,43]. We used these two GFP-tagged proteins as biological indicators to demonstrate our ability to micro-irradiate and analyze in real time cell nuclei responses. We show that XRCC1 is recruited to DNA damage sites a few seconds after irradiation as a function of the total deposited energy and quite independently of the particle LET. RNF8 is recruited to DNA damage sites few minutes after irradiation as a function of the total amount of deposited energy. We observed that the particles LET, and thus the lesion complexity, plays a primary role in driving RNF8 recruitment kinetics.

## 2. Materials and Methods

### 2.1. Beam Line Characteristics

The micro-irradiation setup installed on the AIFIRA facility (*Applications Interdisciplinaires des Faisceaux d’Ions en Région Aquitaine*) is shown in Figure 1. The accelerator (Singletron^TM^, High Voltage Engineering Europa, Amersfoort, The Netherlands) delivers light ion beams with energies up to 3 MeV [21]. This allows cells to be exposed to 3 MeV protons (H^+^) and α-particles (He^+^) presenting a linear energy transfer of 12 and 148 keV·µm^−1^, respectively. As previously described [22], the beam is strongly collimated to reduce the particle flux to a few thousand ions per second on target and focused using a triplet of magnetic quadrupoles to achieve a 1.5 µm beam spot (FWHM). The ion beam is extracted in air and enters the sample through a 4 µm thick polypropylene foil (Goodfellow Cambridge Ltd., Huntingdon, England, cat. no. PP301040) used as a cell support. The ion beam is positioned on the target by means of electrostatic steering. In the case of protons, the mean number of particles (N) hitting the cells was linearly related to the opening time and the relative statistical fluctuation in the number of traversals delivered to the cells decreases as N increases. Considering the mean number of traversals used in this work, this leads to an uncertainty of 10% in the case of 100 protons and 3% in the case of 1000 protons. In the case of α-particles, each particle was detected upstream with a BNCD-coated extraction window (Boron-doped Nano-Crystalline Diamond) from which secondary electrons are collected using a channeltron electron multiplier [15].

The irradiation end-station consists of a motorized inverted fluorescence microscope (Carl Zeiss Micro-Imaging S.A.S, Rueil-Malmaison, France, AxioObserver Z1) equipped with a 14 bits Rolera EM-C^2TM^ Camera (Teledyne Photometrics, Tucson, AZ, USA, cat. no. QImaging) which is positioned horizontally at the end of the beam line. It is equipped with 63× objective (LD Plan-Neofluar 63×/0.75, Optical resolution of 400 nm, Carl Zeiss Micro-Imaging S.A.S, Rueil-Malmaison, France). Fluorescence light is provided by a Light Emitting Diode (LED) illuminating system (Carl Zeiss Micro-Imaging S.A.S, Rueil-Malmaison, France, Colibri2^TM^) with negligible heat production. The image acquisition is performed using the Micromanager software [44].

### 2.2. Dosimetric Simulation Using the Geant4 Monte Carlo Toolkit

We used the Geant4 Monte Carlo toolkit [19], more precisely, its Geant4-DNA processes and models [20,45] available in Geant4 version 11.0. Simulations are performed in a homogeneous liquid water cube of 6 µm in thickness corresponding to the average thickness of a human cell. Energy deposits in the beam spot are calculated using the Livermore electromagnetic physics constructor. For the simulation of track structures, we used a Geant4-DNA Physics list based on the “G4EmDNAPhysics_option4” physics constructor.

### 2.3. Cell Culture, Transfection, and Irradiation

Human osteosarcoma HTB-96^TM^ U-2OS cells (obtained from American Type Culture Collection (ATCC), France) were maintained in McCoy’s 5A medium (Dutscher, Brumath, France, cat. no. L0211-500) supplemented with 10% (*v*/*v*) Bovine Serum (Thermo Fischer Scientific, Illkirch, France, cat. no. 16170-078) and 100 μg·mL^−1^ streptomycin/penicillin (Thermo Fischer Scientific, Illkirch, France, cat. no. 15140-122). All cell lines were kept in an incubator at 37 °C under a 5% (*v*/*v*) CO_2_ humidified atmosphere. A cDNA of human RNF8 inserted into pEGFP-C1 (kindly provided by Jiri Lukas) was used as a construct for stable transfection of GFP-RNF8 [9]. The XRCC1 human cDNA inserted in pEGFP-N1 vectors (kindly provided by Akira Yasui) was used as a construct for stable transfection of GFP-XRCC1 [46]. Viromer Red transfection reagent (Lipocalyx GmbH, Saale, Germany, cat. no. Viromer Red) was used for all transfections, in combination with various expression vectors which were used according to the 125 manufacturers’ guidelines. Transfected cells were plated 48 h after transfection and different geneticin/G418 dilutions from 0.1 to 1 mg·mL^−1^ (Thermo Fischer Scientific, Illkirch, France, cat. no. 10131035) were added 72 h after transfection. After 10 days of drug selection, surviving colonies were checked under fluorescence microscopy and GFP-positive colonies were isolated. Several clones were selected and expanded into cell lines for further analysis. Stably transfected GFP-RNF8 and GFP-XRCC1 cells were platted on the polypropylene surface (Goodfellow Cambridge Ltd., Huntingdon, England, cat. no. PP301040) coated with Cell-Tak^TM^ (Sigma-Aldrich, Saint-Quentin Fallavier, France, cat. no. CLS354240) at a density of 14,000 cells per 20 µL drop, 24 h before irradiation. During microbeam irradiation cells were maintained in FluoroBrite^TM^ DMEM medium (Thermo Fischer Scientific, Illkirch, France, cat. no. A1896701) that ensures a low background fluorescence during images acquisition. Cell nuclei were targeted and irradiated with different absorbed doses of protons or α-particles from 1 to several thousand particles per cell. The protein’s re-localization to the damaged area was followed for 15 and 30 min for GFP-XRCC1 and GFP-RNF8 proteins, respectively.

### 2.4. Image Acquisition, Processing, and Fitting Models

Fluorescence images were acquired online at the microbeam end-station. Time-lapse sequences were chosen to prevent photobleaching and to correct the microscope focus drift on long time periods. Images are recorded at 1 s intervals for 300 s, then at 10 min, 15 min, and 30 min for 100 s each. Figure 2a and Figure 3a show representative time-lapse images acquired after micro-irradiation.

Images were processed using the ImageJ software. Only cell nuclei that did not move during the acquisition were analyzed. The kinetics of GFP-tagged proteins redistribution were measured by recording the fluorescence in irradiated areas The measured values were corrected for non-specific fluorescence bleaching during the repeated image acquisition and were processed as follows:(1)$$Rel\;Int=\frac{\left(I_{\mathrm{foci}}-I_{\mathrm{back}}\right)/(I_{\mathrm{res}}-I_{\mathrm{back}})}{I_{\mathrm{preIR}}}$$
where I_foci_ is the fluorescence intensity in the irradiated area (designated as foci in the following), I_back_ is the background mean fluorescence intensity, I_res_ is the mean fluorescence intensity of irradiated cell nucleus (protein reservoir), and I_preIR_ is the mean of irradiated nucleus fluorescence intensity measured before irradiation.

The redistribution of fluorescence in GFP-RNF8 cells was fitted with a mathematical model for a first-order step response as already described [47]. Data from individual cells were treated individually and fitted with the model:(2)$$Rel\;Int=1+A\left(1-e^{\frac{\left(t-t_{0}\right)}{T}}\right)$$

The distribution of fluorescence in GFP-XRCC1 cells was fitted to a mathematical model proposed by Hable et al. [4].
(3)$$Rel\;Int=1+A\left(1-e^{\frac{\left(t-t_{0}\right)}{T}}\right)*e^{\frac{\left(t-t0\right)}{T_{2}}}$$

The first part of the model function is the same as the one used previously and describes the IRIF formation, in which T represents the mean recruitment time. At the same time, the intensity decreases and it is described by a mean decay time T_2_. A is the highest intensity value reached if a decrease in intensity did not appear. *t*_0_ is the time when focus formation starts. The relevant parameters are *A*, T, and T_2_. These parameters were determined for each cell separately, and then a mean value was calculated. Figure 2b and Figure 3b show representative fits obtained from the cell nucleus from Figure 2a and Figure 3a.

## 3. Results

### 3.1. Energy Deposits and Track Structure of 3 MeV Protons and α-Particles

The charged particles used in this work present different LETs. Indeed, 3 MeV α-particles have a LET of 148 keV·µm^−1^ and 3 MeV protons have a LET of 12 keV·µm^−1^ in liquid water. This lead to a denser spatial distribution of the ionizations and excitations in the case of α-particle irradiations that can be characterized using Geant4-DNA. Figure 4 shows the track structure calculated for a single proton and a single α-particle when passing through 1 µm of liquid water. The amount of energy deposited by one 3 MeV α-particles is about 10 times higher than the one deposited by a 3 MeV proton.

Experimentally, we changed the amount of deposited energy per cell nucleus by irradiating cells with increasing numbers of particles. These charged particles are delivered in a Gaussian distribution of 1.5 µm full width at half maximum (FWHM) to reproduce experimental conditions [15,48,49]. Varying the number and type of incident particles allows for adjusting both the total energy deposited and its distribution at the nanometer scale. As the irradiation is not uniform within the cellular volume, the concept of absorbed dose is of limited use. We chose to calculate the mean deposited energy per spot rather than the absorbed dose as it better relates to the amount of induced molecular damage [50]. This amount of energy deposited is shown in Figure 5. The representation is projected in the X–Y transverse plane and binned with a 0.2 µm step in order to be comparable with the fluorescence images. Figure 5a shows a schematic representation of the beam direction and the geometry used to calculate the deposited energy. The 6 µm depth corresponds to the mean depth of a cell nucleus. Similar mean total deposited energies per beam spot were reached after 10 α-particles or 100 protons and after 100 α-particles or 1000 protons irradiations.

Despite the similar mean total deposited energy per spot by 10 α-particles or 100 protons, it is distributed differently at two scales. First, at the micrometric scale within the beam spot as it can be seen when comparing Figure 5b,c. Even if the total deposited energy is similar for 10 α versus 100 protons and 100 α versus 1000 protons, the maximum energy per pixel is about 2 times higher in the case of α-particles. Second, at the nanometric scale due to the different particle LET as illustrated in Figure 4.

### 3.2. Recruitment Kinetics of GFP-Tagged XRCC1 to DNA Damage Sites after Irradiations

To study the accumulation of GFP-XRCC1 at DNA damage sites, we irradiated GFP-XRCC1 cell nuclei with increasing numbers of protons (100, 500, and 1000) and α-particles (10, 50, 100, and 1000) and we performed online live-cell microscopy. The course of protein kinetics can be described by curves characterized by a recruitment time (T) and a decay time (T_2_), obtained using Equation (3) in the Materials and Methods. The fluctuations of T are small between different cell nuclei while the intensity of fluorescence reached after irradiation (A) and the mean decay time (T_2_) deviate extensively from cell to cell (Appendix A). We focused on the mean recruitment time T and the fluorescence intensity A, which are calculated for at least 16 cell nuclei irradiated during three independent beam times (Appendix A).

Figure 6 shows that T does not vary significantly between 10 and 50 α-particles. When 10 α-particles were delivered, the fluorescent signal was weak and the detection limit of our system was reached. Following irradiations with 50, 100, and 1000 α-particles the recruitment time decreased as a function of the number of delivered particles. In the same manner, a decrease in the mean recruitment time is observed when the number of delivered protons is increased from 100 to 1000. Considering the LET, T is similar when the energy deposited by α-particles and protons is the same. Indeed, the recruitment time after 50 α-particles and 500 protons is the same (the deposited energy per spot is in the same order of magnitude), as the same recruitment time is observed after 100 α-particles and 1000 protons (similar deposited energy).

Although A is varying significantly from cell to cell, this parameter shows a clear correlation with the recruitment time T.

Summarizing these results, we observe that 10 s after irradiation the fluorescence intensity increases in irradiated areas as a function of the number of delivered particles while the recruitment time decreases. This behavior is correlated to the delivered dose independently from the particle type and LET.

### 3.3. Recruitment Kinetics of GFP-Tagged RNF8 to DNA Damage Sites after Irradiations

The ability of GFP-RNF8 protein to accumulate at distinct DNA damage sites was shown previously [15]. In order to monitor the recruitment of GFP-RNF8 at radiation-induced DNA damage sites, images were taken before, during, and up to 30 min after irradiation. To quantify the protein recruitment time, the normalized mean fluorescence intensity of IRIF was plotted as a function of time after irradiation. The protein accumulation follows a curve (Figure 3) characterized by a single time constant (T), obtained using Equation (2), described in the Materials and Methods. The formation of GFP-RNF8 radiation-induced foci became visible within 2 min. The protein reached a steady-state equilibrium 30 min after irradiation. The kinetic parameter T and the fluorescence intensity (A) were calculated after integrating at least 16 cells irradiated over 3 independent beam times (Appendix A) with increasing numbers of protons and α-particles (Figure 7).

By increasing the number from 1 to 10 α-particles, the recruitment time does not significantly change. These measurements can be influenced by the particles’ lateral distribution. By contrast, when 100 α-particles are delivered the recruitment time decreases significantly compared to that obtained after 10 α-particles. When 100 and 1000 protons are delivered, the mean time does not change significantly. Comparing different particles, but the same deposited energy (i.e., 100 α-particles and 1000 protons) the mean recruitment time is shorter when cells are irradiated with α-particles with respect to the mean recruitment time of cells irradiated with protons. This tendency is also observed when cells are irradiated with 10 α-particles and 100 protons.

## 4. Discussion

DNA repair requires a coordinated effort from an array of factors with distinct roles in the DNA damage response. These factors encompass recognition and signaling of DNA breaks, creation of a repair-ready environment, and physical repair of the damage. Given the rapidity of these events, it is imperative to obtain a more comprehensive understanding of the spatio-temporal dynamics of these repair actors within the cell nucleus following DNA damage induction. Live-cell fluorescence microscopy is the preferred method to address this issue [51]. Three approaches are employed for inducing DNA damage in live-cell imaging: genotoxic drugs, endonuclease targeting, and irradiation utilizing different sources. DNA damage induced by irradiation encompasses a wide array of approaches, differing not only in the type of irradiation sources but also in the design of the irradiation scheme in terms of space and time, as well as the potential utilization of chemical sensitizers. IR such as γ-rays, X-rays, or ion beams have been extensively employed to generate DNA lesions, resulting in the formation of IR-induced foci (IRIF) where various repair factors accumulate [4,10,11,14,52,53,54,55,56,57,58]. The types of lesions created primarily consist of SSBs, DSBs, and more complex forms of damage are also observed.

The charged-particle microbeam installed at the AIFIRA facility is fully equipped to perform micro-irradiation experiments and to measure the recruitment kinetics of DNA signaling and repair proteins. We developed and validated the image acquisition and processing methods to allow for a systematic investigation of these phenomena following proton and α-particle irradiations. As a first step, we investigated the radiation-induced behavior of two proteins that respond to DNA damage. Considering the GFP-XRCC1 recruitment time after irradiation by α-particles and protons, different dynamics have been reported for its recruitment and retention at damaged sites following laser-induced damage, depending upon the different wavelengths of the laser light used [16,26,36,46]. Following 365 nm and 405 nm laser irradiation, XRCC1 persists at damage sites. On the contrary, other studies showed that the loss of XRCC1 after irradiation with heavy ions is inconsistent with these data [11,13]. To explain these differences, the authors speculate that the damage produced by 365 nm and 405 nm laser light is highly complex, possibly reflecting a high density of lesions induced. As a consequence, the reparability of such complex DNA damage is reduced, leading to the persistence of XRCC1 at damage sites [6]. Immediate and fast recruitment of XRCC1 is however observed by all authors [11,59,60]. Considering the recruitment time of XRCC1, we found that it is dependent mainly on the deposited energy for both α-particles and protons, with a minimum value of 34 s following irradiation by 1000 α-particles. The measured recruitment times measured in this study are compatible with the ones reported by Liu et al. with higher LET charged particles [61].

Interestingly, the recruitment time of XRCC1 is clearly correlated to the amplitude of the fluorescence intensity in the irradiated spot. Moreover, these parameters only depend on the total energy deposited in the irradiated spot. It does not depend on the particle LET and on the relative number of DSBs and SSBs [62]. This indicates that the recruitment of XRCC1 at damaged sites depends mainly on the total amount of DNA lesions. This is in accordance with the expected role of XRCC1 as a loading platform for other proteins after DNA damage, suggesting it is independent of damage complexity. In addition, we observed that the recruitment of XRCC1 can be up to seven times faster than the RNF8 one. This probably could be in relation to the nature of our particle in terms of energy and LET which produces mainly SSBs (as estimated by Monte Carlo simulation, [62]).

RNF8 is involved in the RNF8-RNF168-BRCA1 molecular complex and has been shown to be a key regulator of DNA repair foci complexes and is mainly involved in the ubiquitination of proteins such as NBS1, γH2AX, BRCA1, or Ku80 [63,64,65,66,67,68,69,70]. We observed that the recruitment of GFP-RNF8 to radiation-induced DNA lesions is dependent firstly on the distribution of ionization. GFP-RNF8 recruitment takes place 1.5 to 4.5 times quicker after α-particle irradiations compared to proton irradiations, while the deposited energy per spot is similar (Figure 7). When 1 or 10 α-particles are delivered the GFP-RNF8 recruitment time does not change but a decrease was observed after 100 α-particles. These responses can be influenced by the microscopic spatial distributions of ionizations. We did not observe changes in GFP-RNF8 recruitment time after increasing the deposited energy with increasing the number of delivered protons. RNF8 behavior induced by charged particles has not been studied extensively yet but studies conducted with laser systems described a strong interaction of RNF8 with MDC1 and NBS1, showing the same recruitment time at DSBs for these proteins [63,66]. This was echoed in a study of MDC1 recruitment time following carbon ion and proton irradiations, which indicated that its recruitment is LET- and absorbed-dose-dependent [4]. In the case of GFP-RNF8, we did not observe a clear correlation of the recruitment time with the fluorescence intensity. Our hypothesis is that the observed behavior of the GFP-RNF8 protein may be affected by the presence of the endogenous RNF8 protein (competition effect), but also by the absence of direct physical interactions with the fragmented DNA strands, or by the potential nature of the primary lesions induced by our type of particles in favor of the production of single-strand breaks.

The results reported in this study illustrate the technical capabilities of the AIFIRA microbeam. In the future, it could be improved by the use of multiple fluorescent dyes enabling to correlate the protein recruitment with the cell cycle phase and/or the chromatin state. Even if limited by the availability of beam times at accelerators, charged-particle micro-irradiation experiments provide quantitative data needed as input to model radiation-induced response and help to understand the relation between dose, radiation-induced effects, and biological responses.

## 5. Conclusions

Charged-particle microbeams coupled to time-lapse fluorescence imaging are relevant tools to study the recruitment kinetics of the proteins involved in DNA repair following exposure to ionizing radiation. Using the microbeam installed at the AIFIRA facility, we measured the accumulation behavior of XRCC1 and RNF8 at damaged sites. This study illustrates the capabilities of our microbeam to trigger localized DNA lesions and follow the subsequent accumulation of DNA repair protein within the first seconds to minutes after irradiation. XRCC1 is recruited to DNA damage sites a few seconds after irradiation as a function of the total deposited energy and quite independently of the particle LET. RNF8 is recruited to DNA damage sites few minutes after irradiation and its recruitment kinetics depends on the particle LET. This first study paves the way for the systematic investigation of various proteins involved in the different DNA repair and signaling pathways.

## Figures and Tables

**Figure 1 biology-12-00921-f001:**
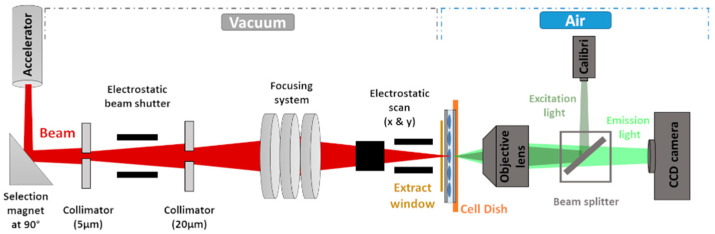
Scheme of the micro-irradiation line setup and microscopy system at AIFIRA. The charged-particle beam (red) is collimated in two stages and then focused in a micrometric spot under vacuum using a triplet of magnetic lenses. The beam is extracted to air through a Si_3_N_4_ window or a thin detector (yellow). The cells are kept in their culture medium and placed vertically in front of the extraction window. Electrostatic scanning plates, placed just before the extraction window, allow the positioning of the beam on the target. A fluorescence microscope (Zeiss AxioObserver Z1) equipped with a CCD camera is placed at the end of the beam line to visualize the sample, locate and target the region of interest, and perform online time-lapse imaging.

**Figure 2 biology-12-00921-f002:**
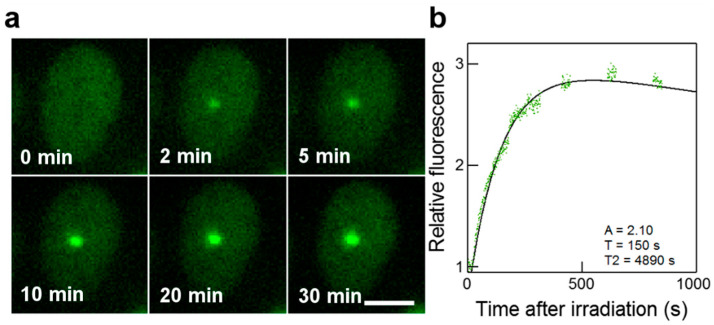
Real-time recruitment of GFP-XRCC1 to the micro-irradiated area. (**a**) Cell nucleus irradiated with 100 α-particles. The time at which the cell is hit by particles is set as t = 0. At t = 0 min, the GFP-XRCC1 protein is distributed homogeneously in the nucleus. Selected time points, covering the signal of the fluorescent spot corresponding to the accumulation of GFP-XRCC1 at the irradiated site are shown. Scale bar: 10 μm. (**b**) Kinetics curve showing the normalized data from the cell nucleus in panel a, fitted using a double-exponential curve and cropped at 15 min.

**Figure 3 biology-12-00921-f003:**
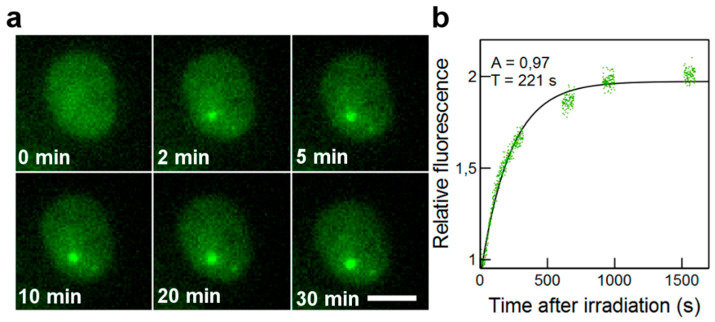
Real-time recruitment of GFP-RNF8 to the micro-irradiated area. (**a**) Cell nucleus irradiated with 100 α-particles. The time at which the cell is hit by particles is set as t = 0. At t = 0 min, the GFP-RNF8 protein is distributed homogeneously in the nucleus. Selected time points, covering the signal of the fluorescent spot corresponding to the accumulation of GFP-RNF8 at the irradiated site are shown. Scale bar: 10 μm. (**b**) Kinetics curve showing the data normalized from the cell nucleus fitted to the first-order model.

**Figure 4 biology-12-00921-f004:**
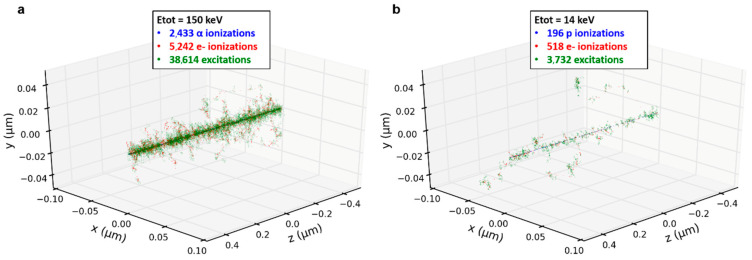
Track structures and number of ionizations of a single 3 MeV α-particle and proton in water obtained with Geant4-DNA. The particles propagate along the *Z*-axis. When (**a**) one α-particle and (**b**) one proton traverse 1 µm of liquid water, the density of ionizations and the deposited energy (Etot) is about ten times higher for α-particles than for protons, due to the α-particle’s higher LET.

**Figure 5 biology-12-00921-f005:**
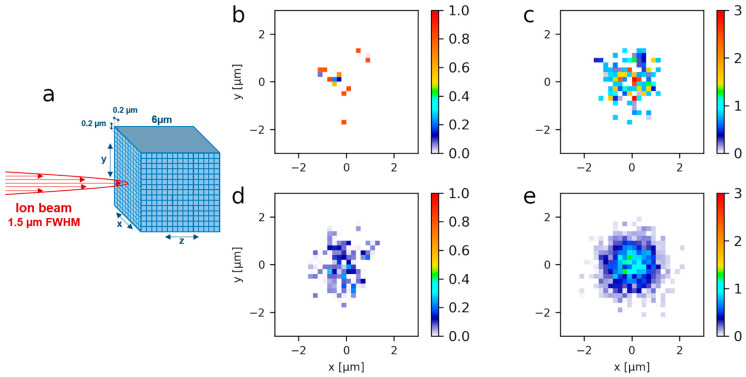
Geant4 simulations of the deposited energy in a simplified cell nucleus. (**a**) Schematic representation of the beam direction (red arrow) and the target (cell nucleus), which is divided into rectangular voxels of 0.2 × 0.2 × 6 µm^3^. The deposited energy is calculated within these volumes. The total deposited energy is similar when (**b**) 10 α-particles and (**d**) 100 protons are delivered, and when (**c**) 100 α-particles and (**e**) 1000 protons are delivered. Color bars indicate the energy deposited per pixel expressed in MeV.

**Figure 6 biology-12-00921-f006:**
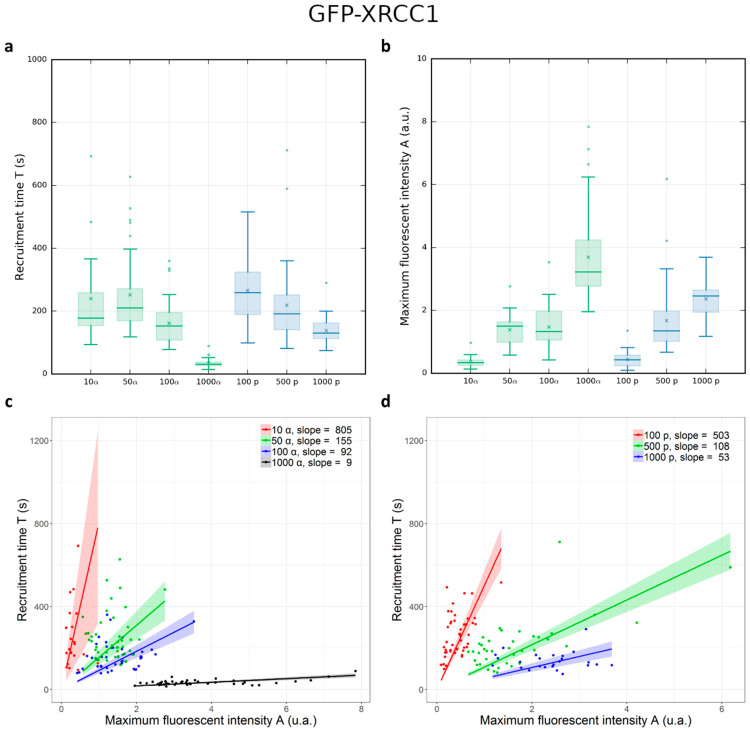
Mean recruitment time (T) and amplitude of the fluorescent intensity (A) of GFP-XRCC1 after proton and α-particle irradiations. (**a**) Box plots representing the distribution of recruitment times (T) of GFP-XRCC1 cells after irradiation with increasing numbers of α-particles (green) and protons (blue). T decreases with increasing numbers of delivered particles independently from their type and LET. (**b**) Box plots representing the amplitude of the GFP-XRCC1 intensity with increasing numbers of α-particles (green) and protons (blue). A = 0 means no increase compared to the fluorescence intensity measured before irradiation. A increases, indicating that the amount of GFP-XRCC1 also increases with the number of delivered particles. Mean and median values are represented as a horizontal line and a cross in the box plots, respectively. Considering the two parameters T and A together for (**c**) α-particles and (**d**) protons, T increases while A decreases when the number of delivered particles increases.

**Figure 7 biology-12-00921-f007:**
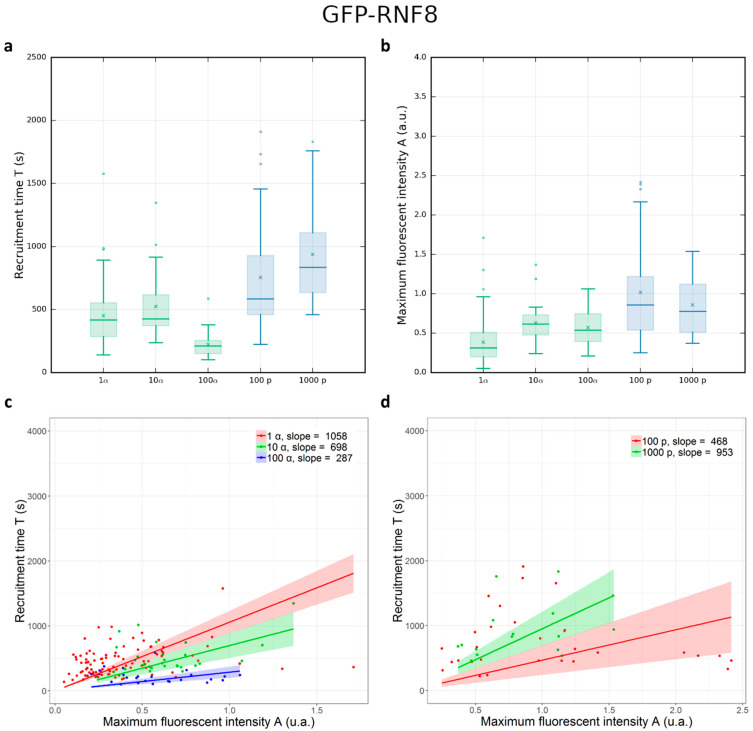
Recruitment time (T) and amplitude of the fluorescent intensity (A) of GFP-RNF8 after proton and α-particle irradiations. (**a**) Box plots representing the recruitment times (T) of GFP-RNF8 cells after irradiation with increasing numbers of α-particles (green) and protons (blue). T changes as a function of the deposited energy when α-particles are increased from 10 to 100; it does not significantly change for the other irradiation condition. GFP-RNF8 protein accumulates faster when cells are irradiated with α-particles with respect to protons. (**b**) Box plots representing the amplitude of the GFP-RNF8 intensity in cell nuclei after irradiation with increasing numbers of α-particles (green) and protons (blue). A = 0 means no increase compared to the fluorescence intensity measured before irradiation. A does not change significantly with the number of delivered particles. Mean and median values are represented as a horizontal line and a cross in the box plots, respectively. Considering the two parameters T and A together for (**c**) α-particles and (**d**) protons, there is no dependency between T and A by increasing the number of delivered particles.

## Data Availability

The data that support the findings of this study are available from the corresponding author, PB, upon reasonable request.

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
