# Peer review of "Recruitment Kinetics of XRCC1 and RNF8 Following MeV Proton and α-Particle Micro-Irradiation"

_biology, 2023, doi:10.3390/biology12070921_

Round 1
Reviewer 1 Report
1. The introduction is very concise. The background needs to contain more information about the proteins being studied and also about the DNA repair process. It lacks information about the behavior or activity of these proteins during the DNA repair process.
2. Why are the RNF8 and XRCC1 assembly kinetics model equation chosen to be different? It seems the XRCC1 model should be used for RNF8 as well since it was first designed for Mdc1.
3. I would suggest a schematic of the experimental setup to be added.
4. How much ionization dose can be tolerated by the cells before it dies completely? Or when other processes get affected?
5. What about proteins like RAD51 or BRCA2, which are one of the main drivers of DSB repair?
Reviewer 2 Report
The manuscript submitted by Muggiolu et al described their investigation on the recruitment kinetics of XRCC1 and RNF8 in living cells after MeV proton and alpha particle irradiation. The cells were irradiated using micrometer scale beam spot and then the fluorescence intensity and its kinetics were measured. The authors found that the mean recruitment time of XRCC1 and the maximum fluorescent intensity is dependent on the irradiated dose, but not ion species; while this is not the case for RNF8 kinetics. This study is very interesting for radiation biology and the DNA repair cascade. The reviewer recommend the publication of the manuscript after the following questions are addressed.
General suggestions:
1.The author should include their main findings in the abstract.
2. how does the measured fluorescence intensity behave before the irradiation? Can the curve presented in Figure 1/2 show data points before the IR to demonstrate the efficacy of the photobleaching compensation?
3. it would be interesting too, to show some mean fluorescence curve measurement of the data of Figure 5ab and Figure 6ab.
4. one can see the relative fluorescence of the accumulated XRCC1 started to decrease after the maximum in Figure 1b, but the dissociation time T2 is not discussed in the manuscript, why? The model given in the reference #43 by W. Liu et al have described the recruitment curve, as well as the recruitment time T and decay T2 with their biochemical meaning in the DNA repair, which might be suitable in interpreting the kinetics difference with different damage complexity.
5. the investigation founds that the mean recruitment time of XRCC1 has strong correlation with the maximum fluorescent intensity. However, with given T and T2, one can obtain the fluorescence maximum from the equation (3) of Line 179. Is there any difference between the calculated maximum and the measured maximum? A detailed analysis of the much-lower T at high dose will be very interesting.
6. line 337, as the measurement did not investigate the fraction of SSB or DSB production, so it is better to remove “ independently of their nature (single or double strand breaks).” Or this point should be extended by furtherer analysis and discussion on the role of RNF8 and XRCC1 related to DSB and SSB.
7. what is the logic behind the sentence between line 360-362? Could the author expands on this point?
8. there are some typos, such as –word connection, special unit symbols, should be checked through the file: f.g line 73, line 85, line 89 in page 2; line 318 of page 9,
Reviewer 3 Report
The article describes a method to detect GFP-labeled DNA repair factors following DNA damage and using microscopy. Attempts to do so were successfully done during the last two decades, and the Introduction and manuscript need to make it clearer what was done earlier, and what is new here.
Specific points.
1. Introduction. It might be helpful for readers to mention the DNA repair pathways where XRCC1 and RNF8 are involved. Simply stating that "These proteins are involved in the recognition, signaling, and repair of DNA lesions." is not sufficient and requires more details.
What is the pathway XRCC1 is involved in (i.e. BER, A-EJ)? Where in these pathways is XRCC1 located and what function does it have? Is it recognition, signaling, or mainly stabilization of the Lig3 enzyme? What are the other members of these pathway(s)?
The same regarding RNF8. What pathway it is part of (ATM-dependent DNA damage response signaling)? Is it directly involved in recognition and repair, or mainly in signaling? What function does it have, and what are the other players in this signaling pathway?
2. Kindly spell key and uncommon abbreviations, e.g. XRCC1, RNF8.
3. Line 70. What is the difference between BER and SSBR? Depending on the definition, BER and NER can be simply parts of SSBR. If another definition is used, it needs to be elaborated.
4. Protons and alpha-particles are only explained in M&M, line 84. Consider defining them earlier, in the introduction, when they are first mentioned.
5. Line 132. Kindly check this sentence "20 l drop, 24 h before irradiation".
6. In Materials and Methods, kindly provide more information regarding the materials/reagents used (producer, country/region/city, catalog number).
7. Line 143. "et 30 min" is probably meant to be "at 30 min"
8. Did anybody publish recruitment of XRCC1-GFP and/or RNF8-GFP to irradiated regions of the nucleus earlier? What methods did they use? What was the dynamics of the process? Why the current study is needed (novelty)?
9. Figure 3. How were these data obtained? Refer to Materials and Methods or published papers, and consider elaborating on how the Figure was generated (which data were used?).
10. Line 239. "The fluctuations of are" seems incomplete.
11. Figure 5. Consider labeling on the Figure that the observations were made for XRCC1-GFP. It is mentioned in the Figure legend, but the Figure itself could be more informative.
12. Make sure the terms are spelled consistently, e.g. Figure-figure-Fig.-fig.
13. Line 318. "ans" probably means "and"
14. Line 318. "fiste" probably means "first"
15. Lines 319-320. "belonging to different DNA damage signaling pathways" is not accurate. While RNF8 is a part of ATM-dependent DNA damage response (DDR) signaling, the XRCC1 is a part of DNA repair pathways (i.e. BER and A-EJ). As suggested above, this should be clarified in the introduction as well.
16. Line 324. "others studies" need to be checked.
17. Lines 341-344. The hypothesis explaining the different speeds of XRCC1 and RNF8 recruitment is nice, however, the readers would expect several alternative points of view, moreover, this statement is not supported by the data in the article, nor by the published data. The reasons for various recruitment speeds could be different, e.g. because the proteins belong to different påathway (DNA repair vs DNA damage response). It would be beneficial to discuss other examples from literature with NHEJ factors being recruited within seconds or milliseconds, while accumulation of gH2AX (the same pathway with RNF8) takes seconds and minutes.
18. Line 347. "DNA ubiquitination of DSBs" looks pretty misleading. Was it really DNA ubiquitination, or is it protein ubiquitination? Which proteins?
19. There is a significant number of typos and inconsistencies to fix.
20. For the recruitment of XRCC1 and RNF8, there is only the appearance of the factors at the irradiated foci shown. Is it possible to detect the dissociation of the factors following DNA repair? How long do these factors stay there? Is the dose too strong that the cells die following irradiation, which prevents DNA repair factors dissociation? Whether the cells are maintained at physiological conditions allowing DNA repair (temperature, nutrients, gases, etc).
Overall, the manuscript focuses on the recruitment of XRCC1 and RNF8 to DNA damage loci. The manuscript at its current stage lacks clear justification of novelty. It can be done by building on the state-of-the-art, which is not sufficiently introduced. The manuscript would benefit from a more detailed and accurate introduction of DNA repair and DNA damage response pathways where XRCC1 and RNF8 belong, respectively. It is important to fix basic biological issues such as "DNA ubiquitination".
There are several typos, missing words, and inconsistencies. The text requires careful proofreading by the authors, potentially using freely available AI-based software.
Reviewer 4 Report
The manuscript titled "Recruitment kinetics of XRCC1 and RNF8 following MeV pro-2 ton and α-particle micro-irradiation" by Muggiolu et al is aimed in demonstrate the potential of microbeam facility equipped with time lapse imaging system to evaluate the recruitment kinetics of XRCC1 and RNF8 after charged particles (alpha and proton) irradiation. The study is well designed and presented with substantial novel thoughts. However, authors need to address following queries/suggestions:
1. Editorial corrections: at a few places like line 89, 132, it looks like micron symbols are missing. At line 143 et 30 min should be 30 min, et should be deleted.
2. For cells U2OS HTB96 tm full source and type of cell (human osteosarcoma) should be mentioned.
3. How the irradiation of particles at the centre of nucleus was ensured as any irradiation at peripheral region of nucleus would show variation in number of DSBs and magnitude of recruitment of XRCC1 and RNF8 proteins
4. For Fig 5 and 6 what is control (sham-irradiated) values? Whether they are before irradiation values? The same should be clarified and mentioned in Figure legends. If any baseline value, the same should be shown in Figures
5. In conclusion major highlight of study i.e. similar repair kinetics if total energy kept same for alpha ad protons and faster repair kinetics of alpha than protons should be highlighted.
6. In Suppl data, Table S1 what is unit of Intensity of fl? Whether for 3 MeV alpha (10 particles) and protons (100 and 500 particles) cells analyzed are from two diff exp. Why there is difference in decrease or no decrease in fl intensity even if number of particles same? The same should be explained?
Round 2
Reviewer 1 Report
Thank you for addressing the concerns.
Author Response
Thank you for your time and effort to review the revised manuscript.
Reviewer 3 Report
The authors revised the manuscript and adressed most of the questions. It is clear to the reviewer now that novelty was not a goal of this study, but the goal was to demonstrate that the technology is available. It is up to Editor to decide on it.
There are two points left unresolved from the first round of the questions though.
1. Original question "1". The involvement of XRCC1 in BER and A-EJ (alternative end-joining) is not introduced. From the point of view of DNA repai field, the current definition provided in the text is not sufficient. SSBR is not a pathway, but a common name of a group of several pathways. It would be necessary to specify which pathway XRCC1 belongs to, and briefly mention the composition of these pathways. Indeed, XRCC1 is known to be a part of BER (single strand break repair) and A-EJ, aka MMEJ (double-strand break repair). This should be stated, ideally with references, even if the authors decide not to describe the entire pathways.
2. Original question 8. On previous publications using the same GFP-tagged proteins. From a point of view of research article, even if novelty is not a goal, it is necessary to briefly mention in the introduction that others used the same proteins to detect the same (or similar) things, although using different settings, different microscopes, different ways to induce DNA breaks, etc. With references. The same in the discussion, it is expected to compare results with the previously published similar results.
The reviewer agrees that there are multiple parameters which are different between the settings and locations (cell types, microscopes, incubators, etc) which makes it hard or impossible to directly measure seconds and minutes of protein recruitment. However, summarizing the previous work and comparing it to results demonstrated in this manuscript, or at minimum providing several known examples, would improve the quality of the report.
3. Finally, if the novelty is not a goal of this manuscript, is it better to choose another type of paper than Article? Technical report? Comment? MDPI has several options to publish, and it is not always that "Article" is the best type of manuscript to decide for.
